# An evolutionary approach to systematic discovery of novel deubiquitinases, applied to *Legionella*

Thomas Hermanns[1], Ilka Woiwode[1], Ricardo FM Guerreiro[1], Robert Vogt[2], Michael Lammers[2], Kay Hofmann[1]

Deubiquitinating enzymes (DUBs) are important regulators of the posttranslational protein ubiquitination system. Mammalian genomes encode about 100 different DUBs, which can be grouped into seven different classes. Members of other DUB classes are found in pathogenic bacteria, which use them to target the host defense. By combining bioinformatical and experimental approaches, we address the question if the known DUB families have a common evolutionary ancestry and share conserved features that set them apart from other proteases. By systematically comparing family-specific hidden Markov models, we uncovered distant relationships between established DUBs and other cysteine protease families. Most DUB families share a conserved aromatic residue linked to the active site, which restricts the cleavage of substrates with side chains at the S2 position, corresponding to Gly-75 in ubiquitin. By applying these criteria to *Legionella pneumophila* ORFs, we identified lpg1621 and lpg1148 as deubiquitinases, characterized their cleavage specificities, and confirmed the importance of the aromatic gatekeeper motif for substrate selection.

## Introduction

Ubiquitination—the posttranslational modification of proteins by covalent attachment of one or more ubiquitin units—regulates a wide variety of cellular processes, including proteostasis, vesicular transport, DNA repair, and the response to pathogens and inflammation ([1], [2]). Ubiquitination is usually targeted to lysine side chains or the amino terminus of the substrate or a substrate-bound ubiquitin unit, giving rise to ubiquitin chains of different linkage types—depending on the lysine residue used for chain elongation ([3]). Deubiquitinating enzymes (DUBs) are involved in multiple steps of the reversible ubiquitin signaling system: on the one hand, they can remove ubiquitin units from substrates or ubiquitin chains, and thus erase the ubiquitin signal. On the other hand, DUBs are required for processing the inactive ubiquitin precursors and are thus indispensable for ubiquitination itself ([4]).

The human genome encodes about 100 different DUBs, which can be grouped into seven classes, based on their sequence relationship and structural fold ([4]). Six of the classes are cysteine proteases: UCH (ubiquitin carboxyl-terminal hydrolases), USP (ubiquitin-specific proteases), OTU (ovarian tumor domain proteins), Josephin (Ataxin-3–like proteins), MINDY (MIU-containing new DUB family), and ZUP1/ZUFSP (zinc-finger ubiquitin protease 1). The six cysteine-DUB classes contain members with ubiquitin peptidase activity (as required for precursor processing and cleavage of linear chains) and ubiquitin-deconjugating isopeptidases, which may be specific for particular substrates or linkage types ([5]). Analogous peptidases and isopeptidases for ubiquitin-like modifiers (UbLs), such as SUMO and NEDD8, also exist. These UbL-proteases are also cysteine proteases, but belong to other families than the true DUBs ([6]). The seventh DUB class, the Jab/MPN domain–associated metalloisopeptidases (JAMM), are often components of large complexes such as the proteasome, where the JAMM protease Rpn11, in conjunction with a catalytically inactive homolog (Rpn8), recycles ubiquitin from degradation targets ([7], [8]). All currently known eukaryotic deubiquitinases can be grouped into these seven DUB-classes, whereas several viruses and bacteria encode effector proteins with DUB activity, which transcend the current DUB classification system ([9]). In fact, most known bacterial deubiquitinases belong to an enzyme class that in eukaryotes is responsible for the processing and deconjugation of the ubiquitin-like modifiers SUMO and NEDD8. Other ubiquitin-fold modifiers, such as UFM1 and ATG8, are cleaved by yet other classes of cysteine proteases, which share with DUBs the property of having isopeptidase activity specific for a glycine residue at the S1 position before the cleavage site ([10]). The presence of deubiquitinase activity in multiple distinct cysteine protease families—and that of related UbL-cleaving activities in other cysteine proteases—raises two interesting questions: First, can we be sure that all DUB classes have already been identified or might there be elusive DUBs hidden among the uncharacterized cysteine proteases? Second, how distinct from each other are the cysteine-DUBs? Did the ubiquitin-isopeptidase specificity really evolve independently many times or are the present-day DUBs rather the divergent descendants of an ancient "proto-DUB" precursor?

[1]Institute for Genetics, University of Cologne, Cologne, Germany   [2]Institute of Biochemistry, Synthetic and Structural Biochemistry, University of Greifswald, Greifswald, Germany

Correspondence: kay.hofmann@uni-koeln.de
Ricardo FM Guerreiro's present address is Institute for Quantitative Genetics and Genomics of Plants, Heinrich-Heine-University Düsseldorf, Düsseldorf, Germany

To address these questions, we performed a bioinformatical analysis of evolutionary relationships between all cysteine protease families annotated in the MEROPS database (11), supplemented with a few other protease families from the recent literature. MEROPS provides a classification of cysteine proteases at two hierarchical levels. At the lower level, proteases with detectable sequence similarity or very similar structures are joined into "families," such as the C19 family, which is mostly equivalent to the USP class of deubiquitinases. Families with a similar 3D-fold and a conserved active site architecture are grouped into "clans." Some of these clans are big and heterogeneous like, for example, the CA clan, which comprises more than half of all cysteine protease families; other clans are much smaller and contain only few families or just a single one. All eukaryotic DUB families belong to the CA clan, whereas SUMO/NEDD8 proteases and some bacterial and viral DUB families belong to the CE clan. By analyzing distant sequence relationships between the families, we obtained a network representation of cysteine proteases, which shows a clear affiliation of bacterial DUBs to eukaryotic DUB or UbL protease families. We next analyzed available DUB structures and DUB family conservation patterns with the aim to identify features shared by multiple DUB classes, but absent from cysteine protease families with other specificities. The only feature with predictive value was the presence of an aromatic residue directly following the catalytic histidine residue. This "aromatic motif" was found to be highly enriched in DUBs and other proteases cleaving after Gly–Gly motifs.

Finally, we attempted a genome-wide prediction of deubiquitinases in the pathogenic bacterium *Legionella pneumophila* by clustering individual ORF-specific hidden Markov models (HMMs) with the HMMs representing the MEROPS families. Overall, 44 *Legionella* ORFs could be placed within the cysteine protease network, 11 of which show an affiliation with a DUB-containing cluster. We selected three interesting examples for experimental characterization of their DUB activity–focusing on "difficult" proteins that are particularly distant from known DUBs but contain the predictive aromatic motif. Two of the selected candidate proteins indeed showed robust DUB activity, whereas the third candidate turned to be an acetyltransferase rather than a DUB—an activity not uncommon for CE clan proteins. These results show that our network- and motif-based approach to DUB prediction is feasible but also indicate that the conserved aromatic motif is not sufficient for discriminating between DUB and acetyltransferase activity.

## Results

### Sequence relationships between MEROPS families

The MEROPS database provides a comprehensively annotated collection of proteases, including isopeptidases and self-processing proteins (11). Since with the exception of JAMM, all known deubiquitinases are cysteine proteases, we focused our analysis on the cysteine protease section of MEROPS v12.0. According to the MEROPS rules, sequence-related proteases are grouped into families and structurally related families into clans. In some cases, families have been divided into subfamilies, which represent deep evolutionary

splits within a family. The rules for deciding on the presence of sequence relatedness are not further detailed, but in most cases, roughly correspond to a significant hit in a BLAST search (12). To assess more distant evolutionary relationships between different MEROPS families, we used HHSEARCH, a software package that implements a HMM-to-HMM comparison of protein families, which is much more sensitive than BLAST-based sequence similarity searches (13). Instead of comparing sequences, this method compares HMMs, which are derived from multiple alignments of entire sequence families. By constructing multiple alignments of each MEROPS family and comparing the resulting HMMs by HHSEARCH, a large number of interfamily similarities could be detected, which are visualized in Fig 1. In this network diagram, each MEROPS family is represented as a box (colored by MEROPS clan affiliation); the thickness of the connecting lines indicates the HHSEARCH *P*-value of the particular similarity. To get a more complete overview of relevant proteases, the cysteine protease families of MEROPS were supplemented by a number of protease families not covered in MEROPS v12.0, including several bacterial DUB families (9, 14), additional CE clan families (15), and the MINDY3/4 family (16); these additional families are represented by yellow boxes. Also included are some MEROPS serine protease families of the mixed clan PA, which are known to be related to cysteine proteases (17).

Some of the connections between MEROPS families are known, a prime example being C64, C65, C85, and C101, which are all considered to be branches of the OTU deubiquitinase family (18). Although the interfamily similarities shown in Fig 1 are typically not detectable by BLAST, more sophisticated bioinformatical analyses of individual DUB families have been performed before, resulting in the publication of some relationships, such as C115 (=MINDY1/2) to MINDY3/4 (16) or the cluster of C54 (=ATG4), C78 (=UFSP), and ZUFSP (19). However, many of the relationships between different MEROPS families are new and unexpected, demonstrating that an HMM-to-HMM–based sequence comparison is useful for adding an intermediate hierarchy level between MEROPS families and clans. A number of observations are of particular interest for our understanding of DUB evolution and predictability: First, almost all viral and bacterial DUB families are connected (and thus related) to established eukaryotic DUB or UbL-protease families. Second, all families within the CE clan are connected, including DUBs, SUMO/NEDD8-proteases, acetyltransferases, and enzymes with ambiguous or unknown specificity (15), suggesting a common ancestry. Moreover, the UCH (C12) and Josephin (C86) deubiquitinase families were found to be related; several other DUB families scored higher with other DUBs than with non-DUB families, but did not reach significance.

### Features shared between DUB families

The sub-significant similarity matches between seemingly unrelated DUB families prompted us to look for conserved sequence features that might be shared between DUB families but are absent from cysteine proteases with other specificities. This sequence-based search was complemented by analyzing the ubiquitin recognition surfaces found in structures of various DUBs co-crystallized with ubiquitin. All DUB classes recognize ubiquitin by at least two regions, one of them formed by the C-terminal tail adjacent to the cleavage site (20), the other one being more

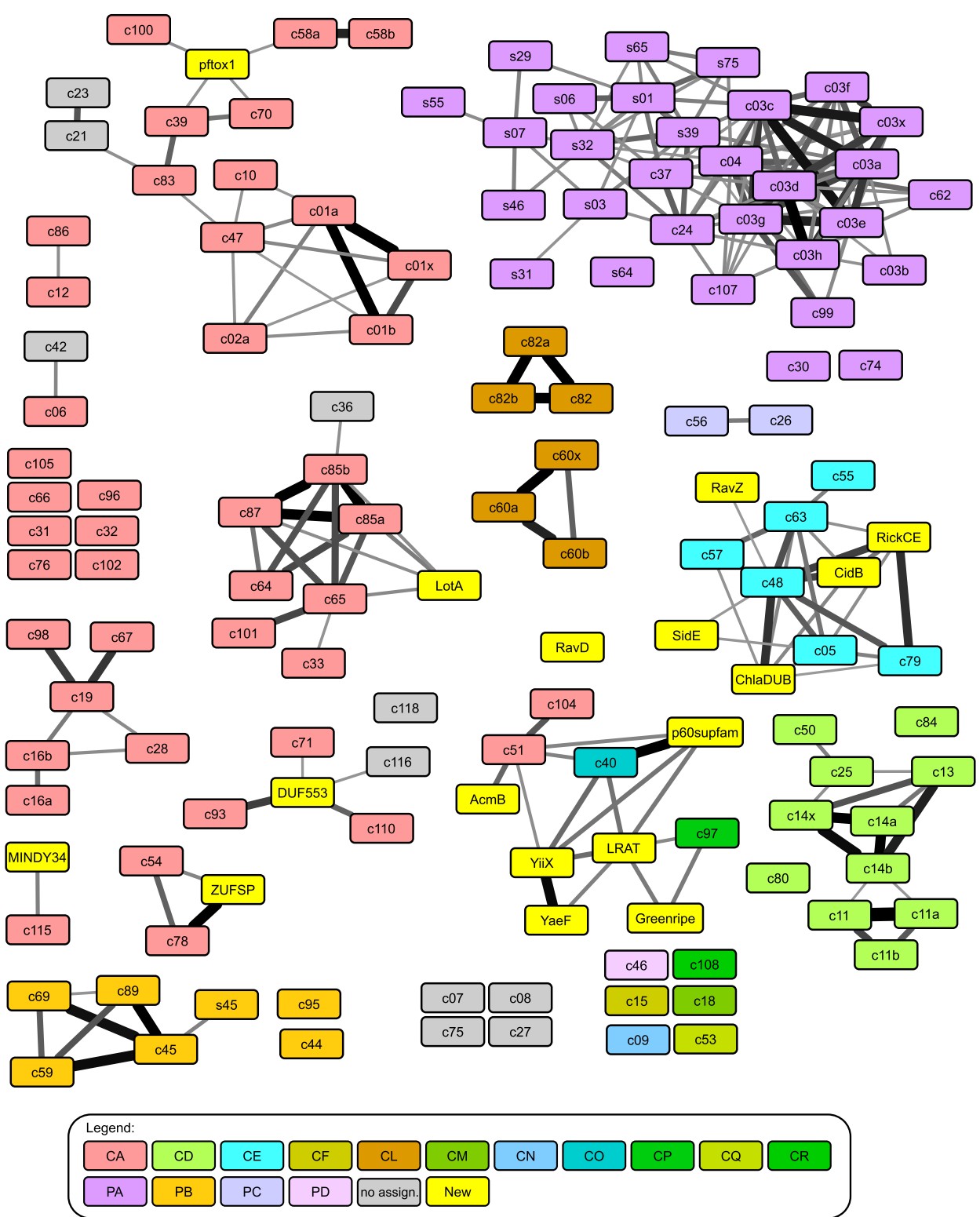

**Figure 1. Sequence relationships network of MEROPS families.**
Colored boxes represent the different protease families, the connecting lines indicate sequence relationships detected by HMM-to-HMM comparisons, ranging from highly significant p < 1E-45 (thickest lines) to the significance threshold of p < 1E-4 (thinnest lines). The boxes are colored according to MEROPS clans, as explained at the bottom of the figure. The labels indicate MEROPS family and subfamily names, the serine proteases of the mixed clans have names starting with "s." Yellow boxes represent additional protease families not yet included in MEROPS v12.0. Grey boxes are MEROPS families not assigned to any clan. Full *P*-value data are provided in Supplemental Data 1. The full cytoscape network used for generating this figure is provided as Supplemental Data 2.

variable. As shown in Fig S1, a common feature of the C-terminal recognition site is the formation of salt bridges between the two arginine residues in the ubiquitin C terminus (**R**L**R**GG) and acidic side chains of the deubiquitinase. In some DUB families such as the USPs (MEROPS C19), the position of the acidic recognition residues is conserved, whereas in other DUBs (such as OTUs or Josephins), different acidic positions are used by different family members. Thus, the presence of suitable acidic residues is difficult to assess in novel DUB candidates without known 3D-structure and hence of little predictive value. The same is true for other ubiquitin-recognition surfaces, which are very heterogeneous even within the individual DUB classes.

More promising is the observation that in most DUB families, the residue directly following the catalytic His residue is aromatic and highly conserved. This His-[Phe|Tyr|Trp] aromatic motif is found both in DUBs of the CA clan with a "Cys-before-His" active site arrangement (USP, UCH, OTU, Josephin, and MINDY) and also in the bacterial DUBs of the CE clan with their inverted "His-before-Cys" active sites. As shown in Fig 2, the conserved aromatic residue in these DUB classes invariably contacts Gly-75 of ubiquitin (RLR**G**G); the aromatic side chain is likely to sterically clash with any side chains at position 75 of ubiquitin—or at least anything bigger than alanine. Thus, the aromatic motif appears to be crucial for restricting substrates to end on Gly–Gly. This prediction is supported by the conservation of the aromatic motif in SUMO/NEDD8 cleaving members of the CE clan, as well as in the USPL1 (C98) family, a SUMO-cleaving relative of the USP deubiquitinases (25). Interestingly, the aromatic motif is absent from proteases cleaving UFM1 (C78) and ATG8 (C54), two modifiers that end on Val–Gly and Phe–Gly, respectively. The only eukaryotic DUB without the conserved aromatic motif is the ZUFSP/ZUP1 family, which is related to the UFM1/ATG8 proteases (19). To quantitatively address the enrichment of the conserved aromatic motif in classical DUBs and proteases for other ubiquitin-like modifiers ending on Gly–Gly, we analyzed the nature and conservation of the residue following the

catalytic histidine in all cysteine protease families present in Fig 1. When considering all families with a universally conserved His-[Phe|Tyr|Trp] motif (found in more than 90% of its members), this motif is enriched 477-fold in canonical DUBs and UbL proteases, as compared with cysteine proteases of other specificities; the significance of this enrichment was p < 3E-19, as evaluated by Fisher's exact test (Table S1). Although the discrimination may not be perfect, the aromatic motif described here is easy to detect in sequence alignments, does not require the availability of structural data, and is likely to help in the identification of divergent DUBs or even novel DUB classes.

## Prediction of *Legionella* cysteine proteases and DUBs

To assess the power of the similarity network approach, combined with filtering for aromatic motif conservation, for the prediction of novel and divergent DUBs, we selected the genome of *L. pneumophila Philadelphia-1* as a test case. As an intracellular bacterium, *L. pneumophila* is known to manipulate the host ubiquitination pathway and several secreted effectors with DUB activity have already been identified (14, 15, 26, 27, 28). To address this question, we first selected promising ORFs from the *Legionella* genome by discarding genes that also exist in extracellular bacteria and are known to have no connections to the ubiquitin pathway. For the remaining 1,202 ORFs, we searched homologs by BLAST, created multiple alignments for each ORF and its homologs, and generated HMMs for subsequent HHSEARCH comparison. Each ORF-centric HMM was compared with all protease-derived HMMs from MEROPS and the additional families from the original clustering shown in Fig 1. The complete result for all 44 *Legionella* ORFs found to be connected to at least one protease family is shown in Fig S2. This finding does not necessarily mean that all 44 *Legionella* ORFs are truly proteases: some MEROPS families include non-proteolytic members such as the transpeptidases of families C82 and C83 or the notorious acetyltransferases of the CE clan family C55.

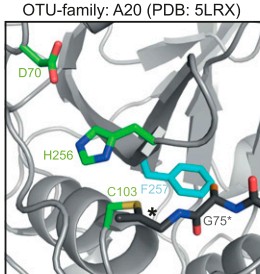

OTU-family: A20 (PDB: 5LRX)

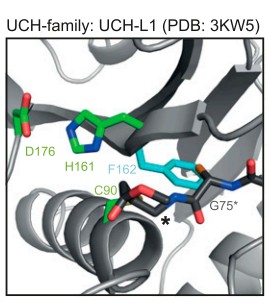

UCH-family: UCH-L1 (PDB: 3KW5)

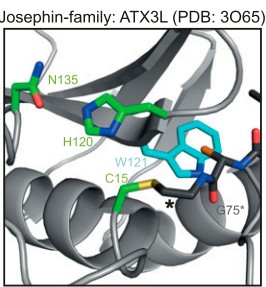

Josephin-family: ATX3L (PDB: 3O65)

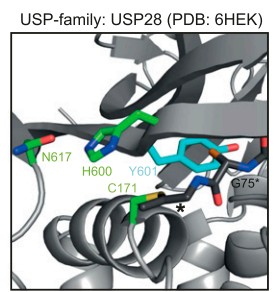

USP-family: USP28 (PDB: 6HEK)

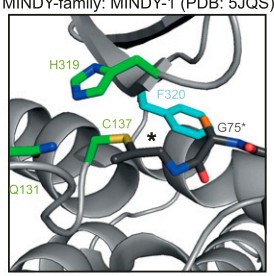

MINDY-family: MINDY-1 (PDB: 5JQS)

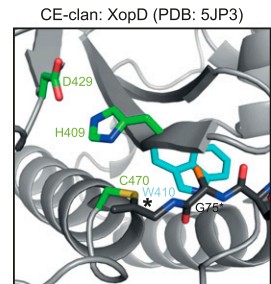

CE-clan: XopD (PDB: 5JP3)

**Figure 2. The gatekeeper role of the conserved aromatic motif.**
The six panels show representatives of the different DUB families (indicated at the top) after reaction with an activity-based Ub probe (the ZUFSP family is not shown because it lacks the aromatic motif) (15, 16, 21, 22, 23, 24). The DUB and ubiquitin structures are shown as cartoon representation and are colored light and dark grey, respectively. The bold asterisk marks the covalent bridge between catalytic cysteine and the ubiquitin C terminus. The conserved aromatic residues of the different DUBs are colored cyan and shown as sticks. In all cases, the aromatic ring system lies adjacent to the penultimate glycine residue of ubiquitin (G75*). To indicate the expected position of a possible side chain at position 75 of the cleaved ubiquitin, a non-physiological $C_\beta$ atom (highlighted in orange) was artificially added to all G75* residues.

Moreover, some of the observed matches might be caused by non-catalytic regions of the proteases or by inactive protease relatives that have lost their catalytic residues.

Because our focus was on the discovery of new deubiquitinases, we analyzed all *Legionella* families with direct or indirect connections to one of the established DUB families in more detail. As shown in Fig 3A, 14 such protein families were identified (indicated by green borders), including all well-characterized *Legionella* DUBs: LotA/lpg2248 (26), the SidE family members SidE/lpg0234, SdeA/lpg2157, SdeB/lpg2156, SdeC/lpg2153 (27), RavD/lpg0160 (14), as well as the ATG8-protease RavZ/lpg1683 (29) and the CE clan acetyltransferase LegCE/lpg2907 (15). Six additional families were identified, two of them (lpg1148 and lpg1949) connected to the CE clan and four (lpg2529, lpg0227, lpg1621, and lpg2952) connected to the OTU cluster within the CA-clan (Fig 3A). Because the *L. pneumophila* member of the lpg2952 family has lost its active site (which is still intact in related bacteria), this protein was not considered as a DUB candidate. The alignments of the OTU and CE clan families in Fig 3B and C show the (limited) conservation of the DUB candidates; it is clearly visible that all of them harbor the conserved aromatic residue after the catalytic His. The only protease without this aromatic motif is RavZ, which cleaves a substrate with a long side chain at the S2-position, obviating the need for the size-restricting aromatic residue.

Lpg2529 and lpg0227 are relatively closely related to LotA and are thus likely to have similar properties. Among the OTU-associated *Legionella* families, we selected lpg1621 for experimental validation because this family is particularly distant from established DUBs—barely exceeding our significance threshold of p < 1E-4. Thus, major structural and functional differences can be expected for this ORF. In addition, we selected the CE clan ORFs lpg1148 and lpg1949 for experimental assessment of their catalytic activity. It should be noted that lpg1949 has the strongest links to YopJ and LegCE, two CE clan acetyltransferases without DUB activity. Nevertheless, lpg1949 contains the aromatic motif, making this ORF an interesting test case for the discrimination between DUB and acetyltransferase activities.

## Initial characterization of the DUB candidates lpg1621, lpg1148, and lpg1949

To validate our bioinformatical predictions, we purified recombinant lpg1621, lpg1148, and lpg1949 proteins from *Escherichia coli* bacteria and tested them for catalytic activity. Because lpg1621 and lpg1148 contain C-terminal transmembrane regions (Fig 4A), C-terminally truncated versions of these two proteins (lpg1621$^{1-348}$ and lpg1148$^{1-306}$) were used; in the case of lpg1949, the full length ORF was expressed. All three candidates were tested for activity against ubiquitin and several ubiquitin-like modifiers by using either model substrates with C-terminally fused fluorogenic AMC (7-amino-4-methylcoumarin) or activity-based probes (30) with a C-terminal alkyne group that reacts covalently with proteases targeting the bond after Gly-76 of ubiquitin (Fig 4B–E). Although the OTU-like protein lpg1621 neither cleaved ubiquitin- nor NEDD8-AMC, it efficiently reacted with Ub-PA, indicating that this protein does indeed have DUB activity (Fig 4B and C). Its C-terminal

cleavage activity appears to be restricted to ubiquitin because there was only a negligible reaction with NEDD8-PA and a complete lack of reactivity toward ISG15-, SUMO1-, and SUMO3-PA (Fig 4C).

The CE clan ORF lpg1148 has previously been proposed to be a metaeffector, protecting cells from the toxic effects of the unrelated effector LegC3 by acting as a deubiquitinase (28). However, the ubiquitin chain–degrading activity and linkage preference of lpg1148 (=LupA) were not characterized in that study. We found lpg1148/LupA to efficiently cleave ubiquitin-AMC, whereas NEDD8-AMC was not cleaved at all (Fig 4B). In line with these results, LupA did only react with the activity-based ubiquitin probe Ub-PA but was inert toward all other tested UbL-PA probes (Fig 4D), confirming the specificity of lpg1148/LupA for ubiquitin.

Lpg1949 is a CE clan member related closer to the acetyltransferase families YopJ and LegCE than to the ubiquitin-processing members of the clan. However, a recent study identified another CE clan member ChlaDUB1 to have a dual activity as acetyltransferase and deubiquitinase—using the same active site for both activities (31). Because lpg1949 also contains the auspicious aromatic motif, we tested it for DUB activity. However, lpg1949 neither cleaved the tested AMC substrates nor did it react with any of the activity-based probes, suggesting that lpg1949 is not a DUB (Fig 4B and E). Instead, lpg1949 shows activity as an acetyltransferase: the incubation of full length lpg1949 with acetyl-CoA and the model substrate MEK6 led to the formation of both auto-acetylated enzyme and acetylated MEK6, as visualized by a Western blot using an antibody-specific for acetylated lysine residues (Fig 4F). Analogous products were also formed using the truncated version lpg1949$^{1-291}$.

## Linkage specificity of lpg1621 and lpg1148

In humans, OTU type DUBs are highly selective for certain chain types (18). To investigate whether lpg1621—despite its great divergence from known OTU proteins—also shows such a preference, we tested its activity against a panel of differently linked di-ubiquitin species. As shown in Fig 5A, lpg1621$^{1-348}$ efficiently cleaved K63-linked di-ubiquitin, whereas the other chain types were not cleaved at all. This specificity was confirmed by cleavage of longer Ub$_4$ chains. Again, only K63-linked chains were cleaved (Fig 5B), indicating that lpg1621 is—like the eukaryotic OTU proteases—a highly specific DUB. In line with the bioinformatical prediction (Fig 3C), the cleavage of K63-linked di-ubiquitin is completely abrogated by mutation of Cys-29 to an alanine residue, indicating that Cys-29 is the catalytic cysteine of this OTU-like protease (Fig 5C).

Typical bacterial CE clan DUBs are less linkage specific and cleave multiple chain types, often with a slight preference for K63-linked chains (9). When tested against a panel of different di-ubiquitin species, lpg1148/LupA cleaved K11, K29, K48, and K63 chains with comparable efficiency. K6 and K33 were cleaved more slowly, whereas linear chains were not cleaved at all (Fig 5D). A previous publication on the metaeffector role of lpg1148/LupA suggests a catalytic core consisting of amino acids 123–305 and provides a crystal structure of this region (28). However, our data suggest that this region is not sufficient for DUB activity. As shown in Fig 5E and F, the lpg1148$^{123-305}$ construct is as inactive against K48-linked di-ubiquitin and ubiquitin-AMC as the catalytic cysteine

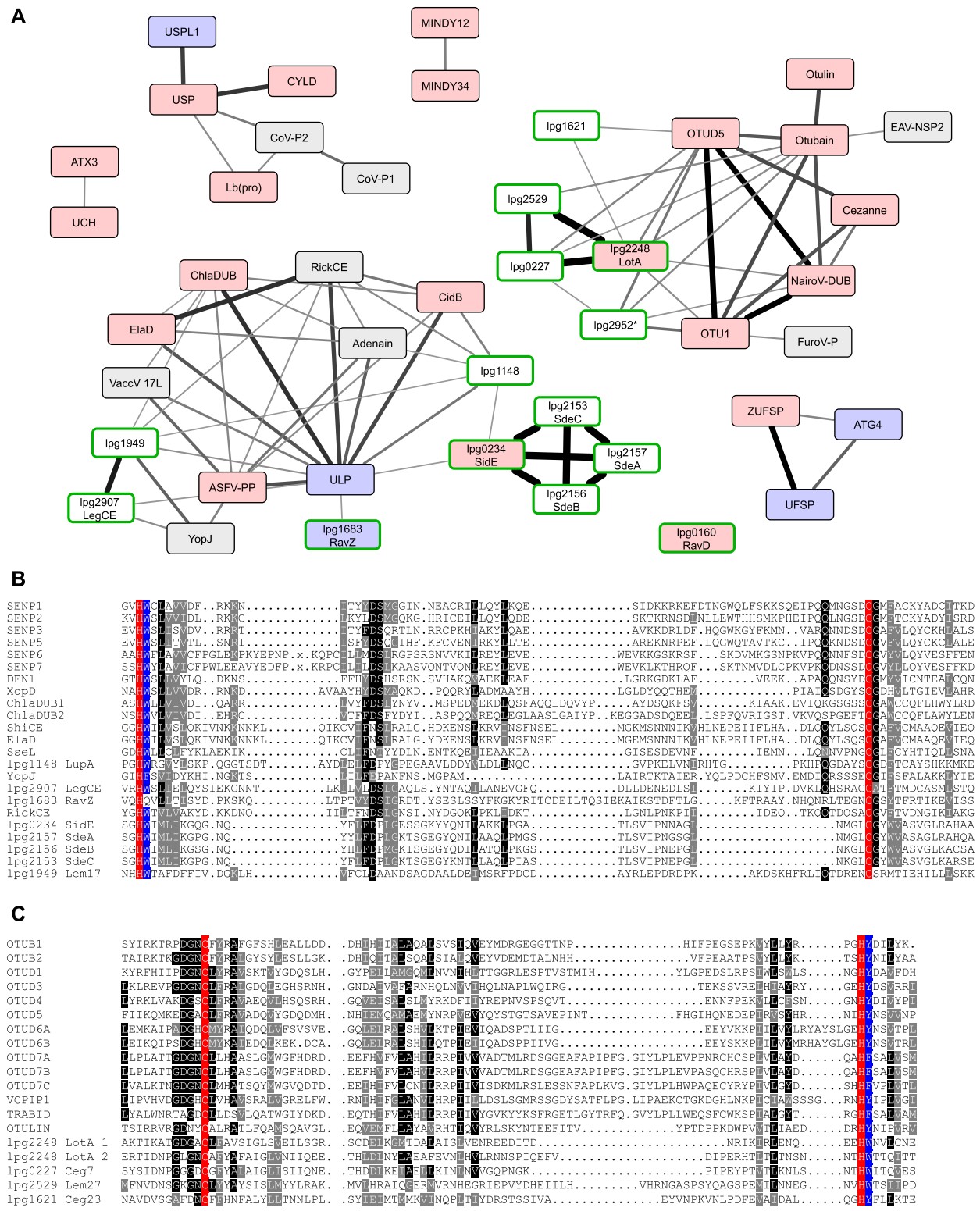

**Figure 3. Discovery of new DUB families in the *Legionella* genome.**
**(A)** Subnetwork of Fig S2, containing known DUBs and DUB candidates. The network was generated from the MEROPS network as shown in Fig 1 by adding all significant links to *Legionella* single-ORF families. Red and blue boxes indicate known DUBs and UbL-proteases, respectively. Boxes with green borders indicate *Legionella* families. The thickness of the connecting lines indicates the significance of family-to-family relatedness, analogous to Fig 1. Full *P*-value data are provided in Supplemental Data 1. The full cytoscape network used for generating Fig 3A is provided as Supplemental Data 3. **(B, C)** Alignment of the CE clan (B) and OTU (C) catalytic core regions. The newly identified *Legionella* proteins are compared with representative human family members and known bacterial DUBs (alignment adapted from reference 9). Residues invariable or conservatively replaced in at least 50% of the sequences are indicated by black and grey background, respectively. Catalytic Cys and His residues are highlighted in red and the conserved aromatic "gatekeeper" residue is highlighted in blue.

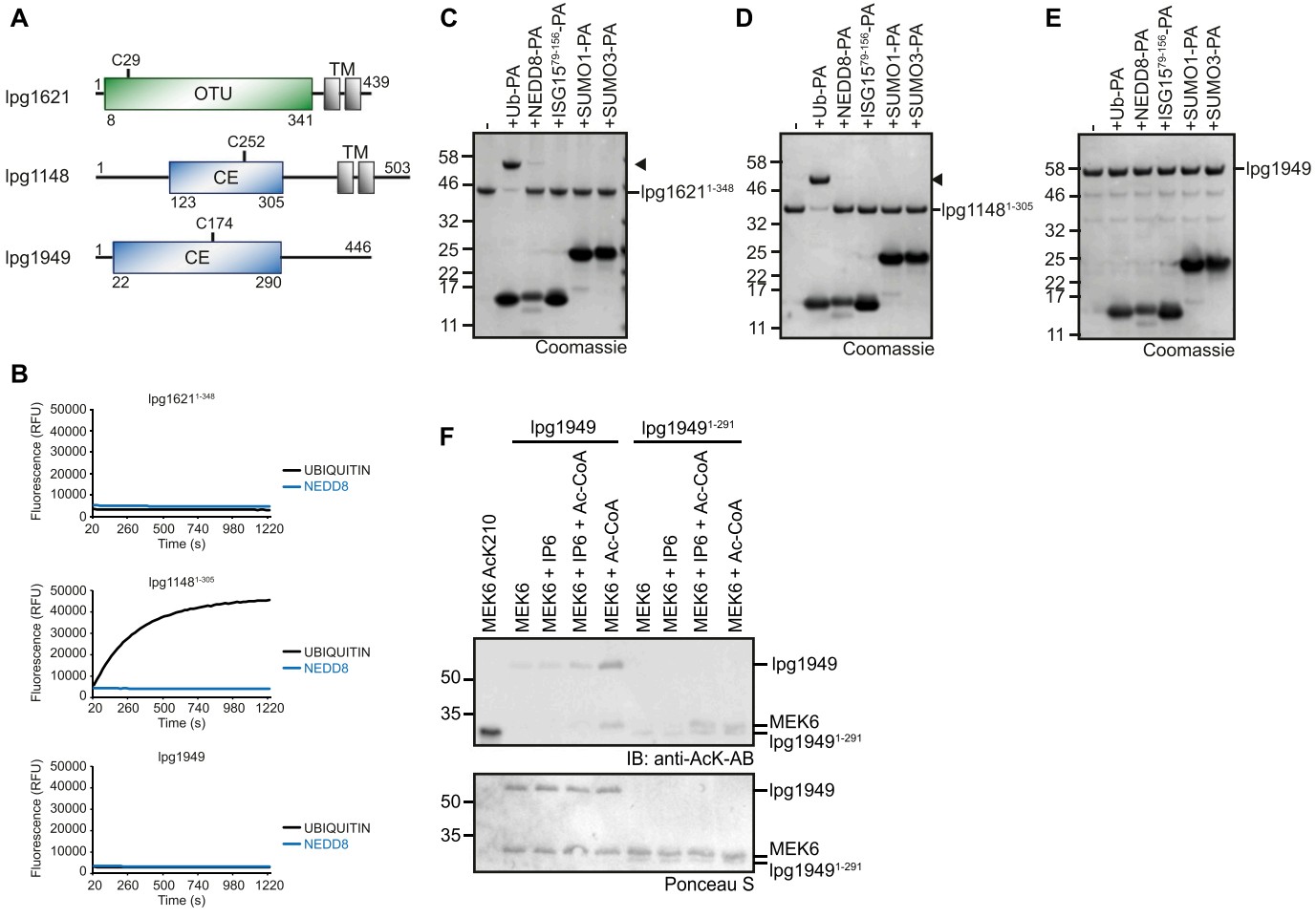

**Figure 4. Lpg1621 and lpg1148 are active deubiquitinases.**
**(A)** Domain scheme of novel deubiquitinase candidates. Protease domains are depicted in green (OTU) or blue (CE), whereas transmembrane domains (TM) are shown in black. **(B)** Activity assays with ubiquitin and NEDD8-AMC substrates shown as released fluorescence (RFU) over time (s) with lpg1621[1–348], lpg1148[1–305], or lpg1949. Shown RFU values are the means of triplicates. **(C, D, E)** Activity-based probe reaction of lpg1621[1–348] (C), lpg1148[1–305] (D), or lpg1949 (E) with Ub- and UbL-activity–based probes. Black arrowheads mark the shifted band after reaction. **(F)** Acetyltransferase activity test performed with full-length lpg1949 or lpg1949[1–291] and MEK6 as a substrate. Acetylated MEK6 (AcK210) was used as a positive control. The assay was performed with and without inositol-hexaphosphate (IP6), an activator of bacterial acetyltransferases ([45]). Acetylation was visualized by immunoblotting with anti–AcK-antibody and Ponceau S staining served as a loading control.

mutant C29A (Fig 5E and F). This observation is in line with published data on other CE clan proteases, such as SseL or XopD ([15]). These CE clan DUBs recognize the substrate ubiquitin by a variable-region 1, which resides N-terminally of the catalytic core domain and is crucial for activity. To map the putative variable region 1 region of lpg1148/LupA, we tested various N-terminal truncations and found that the entire N-terminal region is needed for DUB activity (Fig 5G and H).

## Role of the aromatic motif in substrate selection

The structural analysis shown in Fig 2 suggests that the conserved aromatic motif interacts with Gly-75 in ubiquitin or corresponding residues occupying the S2 site during the cleavage of SUMO and NEDD8. The recently determined crystal structures of lpg1148 and lpg1621 ([28], [32]) confirm the expected orientation of the aromatic motif. Both structures cover only the protease without bound

ubiquitin substrate. Structural superposition with related proteases that include a covalently linked ubiquitin suggest that in lpg1148 and lpg1621, the aromatic side chain also contacts Gly-75 of the outgoing ubiquitin (Fig S3A and B).

However, at this point, it is not clear if the aromatic residue just prevents larger side chains from accessing the S2 site, or if it also makes positive contributions to the recognition of glycine. To address this question, we mutated the aromatic motifs of lpg1621 and lpg1148 to alanine and tested the influence of these mutations on the cleavage of wild-type ubiquitin and ubiquitin mutants with larger side chains at position 75. For the CE clan DUB lpg1148/LupA, the W184A mutant lost the ability to process ubiquitin-AMC and showed drastically reduced cleavage of wt K48-linked di-ubiquitin (Fig 6A and B). Similarly, the Y271A mutant of the OTU-type DUB lpg1621 showed markedly reduced activity against wt K63-linked ubiquitin chains (Figs 6C and S3C). When comparing activity-based probes made from ubiquitin with gradually increasing side chains

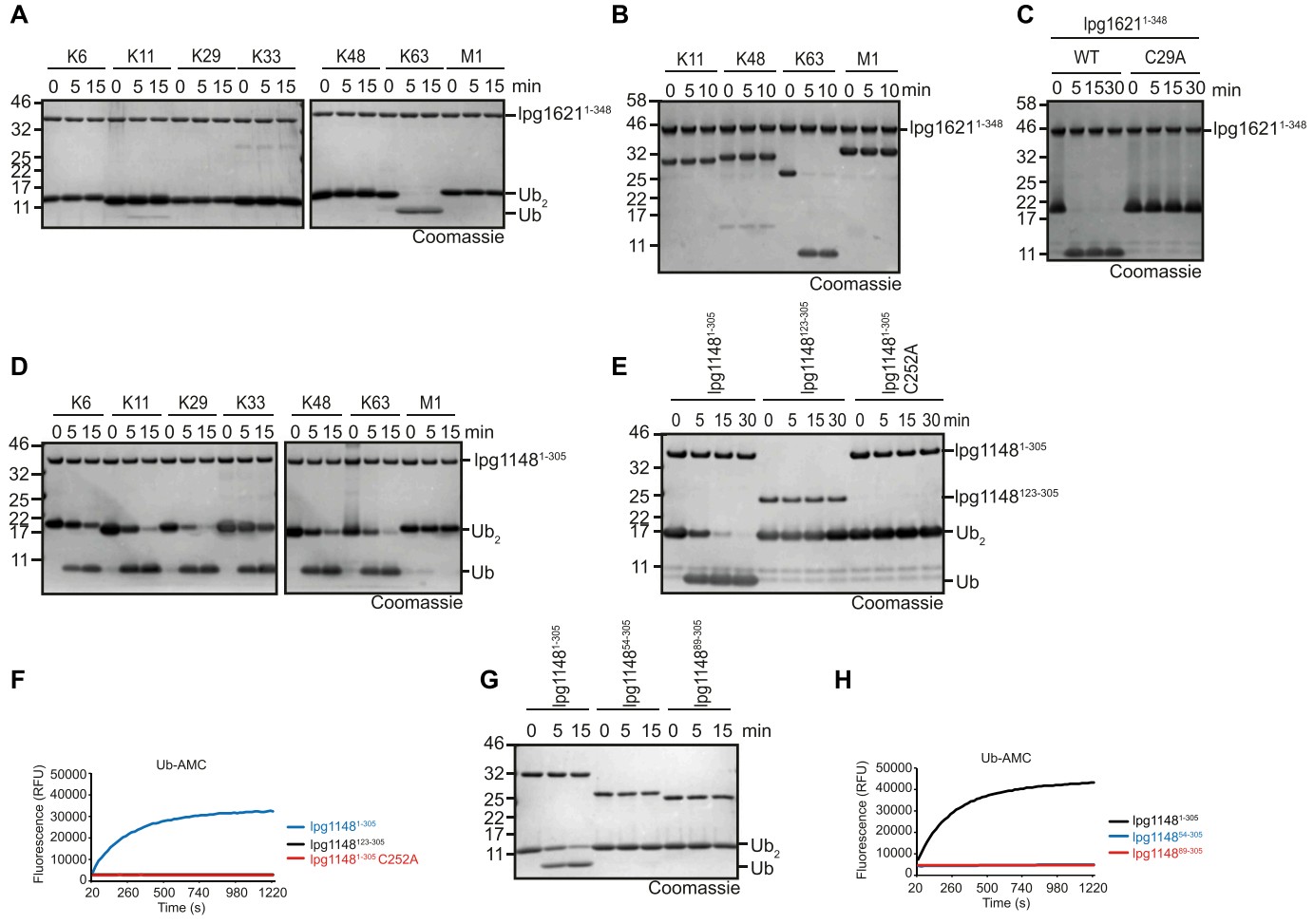

**Figure 5. Lpg1621 cleaves only K63-linkages, lpg1148 is more promiscuous.**
**(A, B)** Linkage specificity analysis with lpg1621[1–348]. **(A, B)** A panel of di-ubiquitin (A) or tetra-ubiquitin (B) chains was treated with lpg1621[1–348] for the indicated time points. Cleavage was analyzed by SDS–PAGE and Coomassie staining. **(C)** Activity assay of lpg1621[1–348] or the catalytic inactive mutant lpg1621[1–348] C29A against K63-linked Ub₂. **(D)** Linkage specificity analysis with lpg1148[1–305] performed as described in (A). **(E, F)** Activity assays of lpg1148[1–305], an N-terminal truncation harboring only the catalytic domain (lpg1148[123–305]) or the catalytic inactive mutant lpg1148[1–305] C252A against K48-linked Ub₂ (E) or ubiquitin-AMC (F). **(G, H)** Activity assays of lpg1148[1–305] or two N-terminal truncations (lpg1148[54–305]/lpg1148[89–305]) lacking a putative variable-region 1 region against K48-linked Ub₂ (G) or ubiquitin-AMC (H).

at position 75 (wt, G75A, and G75V), an interesting trend became visible: The wild-type versions of both lpg1148 and lpg1621 reacted equally well with the wild-type probe and the G75A probe but did not react at all with the larger G75V probe. By contrast, the aromatic motif mutants of the two *Legionella* DUBs showed reduced reactivity with the wild-type probe, little change in reactivity with the G75A probe but an increased reactivity against the G75V probe (Fig 6D and E). These findings strongly support a dual role for the aromatic motif in stabilizing the binding to Gly-75 and preventing a recognition of Val-75 and probably also larger side chains.

To investigate whether these roles are conserved within other DUB families as well, representative candidates from three abundant DUB families were chosen and the respective aromatic residue mutated to alanine. In the case of USP21, both the expected catalytic and gatekeeper roles of Tyr-518 could be confirmed (Fig 6F). In the case of the ataxin-3–like protease JOSD2, the aromatic Trp-126 residue appears to be essential for catalysis or structural integrity: whereas the wild-type enzyme reacted with the wild-type

Ub-PA and–less so–with the G75A probe, no reaction was observed with the G75V probe, suggesting that there is a gatekeeper (Fig 6G). However, the W126A mutation of JOSD2 totally abolished all reactivity and thus did not allow the evaluation of a possible gatekeeper role of this residue. By contrast, the UCH family member UCHL3 appears to be more permissive to changes at the G75 of ubiquitin: The wild-type enzyme reacts readily with the wild-type and G75A probes and even shows some reactivity with the G75V probe (Fig 6H). The aromatic F170A mutation shows only a very mild increase in G75V reactivity.

# Discussion

The presence of a conserved aromatic residue following the catalytic histidine is a prominent feature of almost all classes of deubiquitinases and UbL-proteases, at least those that cleave the

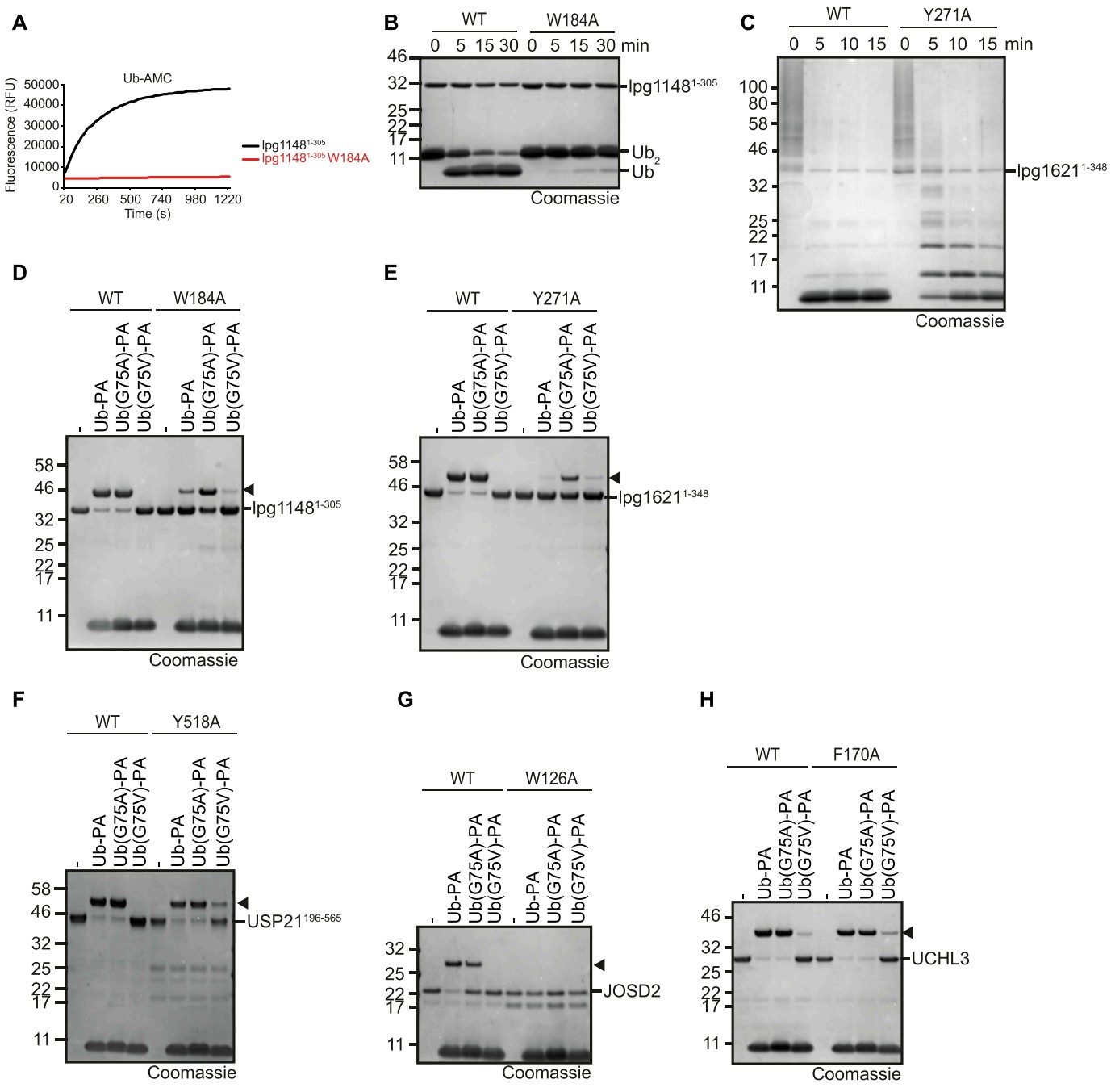

**Figure 6. The aromatic motif has both stabilizing and gatekeeper functions.**
**(A, B)** Activity assays of lpg1148[1–305] or the aromatic motif mutant lpg1148[1–305] W184A against ubiquitin-AMC (A) or K48-linked $Ub_2$ (B). **(C)** Activity assays of lpg1621[1–348] or the aromatic motif mutant lpg1621[1–348] Y271A against K63-linked $Ub_{6+}$. **(D, E, F, G, H)** Activity based probe reaction of lpg1148[1–305] (D), lpg1621[1–348] (E), USP21[196–565] (F), JOSD2 (G), or UCHL3 (H) with Ub-PA. WT or aromatic motif mutants of both DUBs were tested against WT, G75A, or G75V Ub-PA. Black arrowheads mark the shifted band after reaction.

modifier after the canonical Gly–Gly motif. By contrast, most other classes of cysteine proteases lack this feature, although many protease families contain sporadic members with an aromatic residue at this position. However, in the absence of sequence conservation, such occurrences are most likely caused by chance alone and not indicative of a DUB-related activity. The experiments shown in Fig 6 show that for several DUB classes, the aromatic motif has a role in substrate selection by preventing residues with large side chains to position: the tested members of the OTU, USP, UCH, Josephin, and CE clan class react well with an activity-based probe based on wild-type ubiquitin, tolerate (to some degree) a G75A mutated version, but do not react with a G75V-mutated probe. An

exception is UCHL3, which shows a modest reactivity toward the G75V probe. These results are in good agreement with a previous study on the reactivity of USP, UCH, and OTU representatives against a panel of tetrapeptide substrates (33). Interestingly, when mutating the aromatic motif of our OTU, USP, UCH, and CE clan examples to alanine (Fig 6), we observed an increased reactivity against the G75V probe, strongly supporting the P2 "gatekeeper" role of the aromatic motif. The loss of activity against the wild-type ubiquitin probe, which is most drastic for the aromatic motif mutant of JOSD2 but also visible in other DUB classes, hints toward an additional role of the aromatic residue in promoting catalysis.

The strong conservation of the aromatic motif in DUBs and GG-cleaving UbL proteases raises interesting questions about the evolution of these proteases. Are all DUBs monophyletic with the aromatic motif as a conserved ancestral feature or does the aromatic motif in different DUB classes constitute a case of homoplasy? These questions cannot be answered satisfactorily with the currently available data. Structurally, all known DUBs more or less closely follow the papain fold, which already narrows down their evolutionary origins (34). The papain fold, which in MEROPS is typically represented by the CA-clan, is also found in the CE clan as a circularly permuted form; several recently identified OTU-type bacterial deubiquitinases also show circular rearrangement of the core structural elements (35). However, the papain fold is also seen in many non-DUB proteases and our HMM-to-HMM comparisons shown in Figs 1 and 3A reveal significant sequence relationships only between some DUB and UbL-protease families such as ATX3-UCH or ZUFSP-UFSP-ATG4, leaving other families unconnected. We can only speculate that an improvement of the underlying HMMs might uncover more such relationships. Interestingly, conserved aromatic motifs are also found in two cysteine protease families without a link to ubiquitin- or UbL cleavage (Table S1). One family (C39, bacteriocin-processing peptidase) appears to target Gly–Gly–based cleavage sites in linear peptides, for the other one (C57, vaccinia virus I7L processing peptidase), a substrate has not yet been described. At this point, several evolutionary scenarios remain possible, either with or without an ancestral status of the aromatic motif.

Irrespective of the detailed evolutionary history, our data demonstrate the feasibility of a bioinformatical DUB prediction based on i) the presence of (distant) similarity to established DUBs or UbL proteases and ii) the conservation of the aromatic motif after the catalytic histidine. Not only did this approach allow the retrospective "discovery" of nearly all known bacterial DUBs, it also correctly predicted the DUB activity of lpg1148/LupA and lpg1621, which were unknown to us at the time of the analysis. Recently, a deubiquitinase activity for Ceg23 (=lpg1621) was published (32), with catalytic properties in full agreement to those reported here (Fig 5). On the other hand, a DUB prediction relying only on these two criteria is not perfect, as highlighted by two problematic cases: the lack of RavD detection and the discrimination of DUB/acetyltransferase activity in CE clan enzymes.

RavD, a recently described *Legionella* DUB specific for linear chains (14) was found in our screen only because a RavD-HMM was already present in the query set—no significant link to any other DUB class was detected (Fig S2). To assess the possibility of other *Legionella* DUBs eluding our screen, we investigated the RavD

situation in detail. First and foremost, RavD is an extremely difficult case in several respects. The published structure of RavD resembles the CA-clan fold but is otherwise rather singular (14); structure comparison programs such as Dali (36) report only similarities at Z-scores below six, with the non-DUB cysteine protease AvrPphB as the best hit. Moreover, RavD-like proteins are very rare and are absent even from most closely related *Legionella* species. As a consequence, the HMM for lpg0160/RavD is based on very few and nearly identical sequences, two factors that severely limit the detection sensitivity of HMM-to-HMM searches. It can be expected that similar cases—very distant DUBs with few available sequences—might also fail on the similarity criterion. However, RavD does possess the conserved aromatic motif, which could serve as a component in similarity-independent DUB predictions.

A different problem is encountered with lpg1949, a *Legionella* ORF that clearly clusters with the CE clan and contains a conserved aromatic motif—thus fulfilling both criteria of our screen. The CE clan is known to include both DUBs and acetyltransferases, with both activities using the same active site residues (15). There is at least one enzyme (ChlaDUB1) that combines both activities (31). The tight clustering of the lpg1949 family with YopJ and LegCE, two known acetyltransferases, already raised our suspicion that lpg1949 might also be an acetyltransferase. Because the aromatic tryptophan residue following the catalytic histidine has no known role in the acetyltransferase reaction, its conservation has been proposed to indicate the presence of DUB activity (15). However, our results (Fig 4B, E, and F) confirm that lpg1949 is an acetyltransferase rather than a DUB, suggesting that the aromatic position is not useful for discriminating the enzymatic activities within the CE clan. Despite these reservations, we are convinced that both the similarity clustering criterion and the conservation of the His-[Phe|Tyr|Trp] aromatic motif will prove instrumental for the future discovery of novel DUBs in host and pathogen genomes.

## Materials and Methods

### MEROPS similarity network

A nonredundant set of protease active domain sequences, corresponding to all MEROPS v12.0 (11) protease families, was retrieved from the site https://www.ebi.ac.uk/merops in February 2019. Only sequences belonging to the cysteine protease clans or to mixed clans involving cysteine proteases were used for the subsequent analyses. In sequence families with more than 300 members, highly similar entries were removed using the CD-HIT software (37). The collection of MEROPS-derived protease families was supplemented by several recently published families of bacterial DUBs and related proteases (9, 14, 15, 16). For each family, the protease sequences were aligned using the L-INS-I algorithm of the MAFFT package (38) and the multiple alignments used for the generation of HMMs, using the HHSEARCH software package (13). Finally, all family-specific HMMs were compared against each other using HHSEARCH (13) with a significance cutoff of p < 1E-4. The results were imported into Cytoscape (39), clan and activity annotations were added manually, and the resulting network was exported and used as the basis for Fig 1.

## Identification of *Legionella* DUB candidates

Sequences of *Legionella Pneumophila Philadelphia-1* ORFs and other bacteria were obtained from the proGenomes2 resource (http://progenomes.embl.de). ORFs with significant homology to proteins from the nonpathogenic *Bacillus subtilis* are unlikely to encode DUBs and were not further considered. For the remaining ORFs, homologs were searched by BLAST, and the resulting families were aligned using the L-INS-I algorithm of the MAFFT package (38). If necessary, family size was reduced before alignment by removal of highly similar proteins (37). After generating HMMs from the ORF-specific family alignments, the HMMs were compared by HHSEARCH against the protease-specific HMM collection described above (13), using a significance cutoff of p < 1E-4. The results were imported into Cytoscape (39) and the resulting network was exported and used as the basis for Figs 2 and S2. All *Legionella* families with connections to DUB-containing protease clusters were manually checked for active site conservation and presence of an aromatic residue following the active-site histidine. Only families fulfilling both criteria were considered for experimental validation.

## Cloning and mutagenesis

The *Legionella* DUB candidates lpg1148, lpg1621, and lpg1949 were amplified by PCR from genomic DNA (kind gift from A. Hamprecht, University Hospital of Cologne) using the Phusion High Fidelity Kit (New England Biolabs). JOSD2 and UCHL3 were amplified from HEK293-derived cDNA accordingly. The amplified genes were cloned in the pOPIN-S vector (40) using the In-Fusion HD Cloning Kit (Takara Clontech). Point mutations were generated using the QuikChange Lightning kit (Agilent Technologies). Constructs for ubiquitin-PA purification (pTXB1-ubiquitin$^{1-75}$) and pOPIN-S USP21$^{196-565}$ were a kind gift of D. Komander (WEHI). NEDD8, ISG15$^{79-156}$, SUMO1, and SUMO3 were amplified from human cDNA and cloned in the pTXB1 vector (New England Biolabs) by restriction cloning according to the manufacturers protocol.

## Protein expression and purification

All *Legionella* DUB candidates, including all truncations and mutants were expressed from pOPIN-S vector with an N-terminal 6His-Smt3-tag. The truncated construct LPG1949 encompassing residues 1–291 (LPG1949$_{1-291}$) was expressed as GST-fusion protein using the vector pGEX-4T5/TEV, a vector derived from pGEX-4T1 (GE Healthcare). MEK6 and acetylated MEK6 AcK210 were expressed from pRSF-Duet1 as N-terminal 6His-tagged fusion proteins.

All constructs expressing DUBs were transformed into *E. coli* (Strain: Rosetta (DE3)pLysS), and 2-6 l cultures were grown in LB medium at 37°C until the OD$_{600}$ of 0.8 was reached. The cultures were cooled down to 18°C, and protein expression was induced by addition of 0.2 mM isopropyl $\beta$-d-1-thiogalactopyranoside (IPTG). After 16 h, the cultures were harvested by centrifugation at 5,000$g$ for 15 min. After freeze thaw, the pellets were resuspended in binding buffer (300 mM NaCl, 20 mM Tris, pH 7, 20 mM imidazole, and 2 mM $\beta$-mercaptoethanol) containing DNase and lysozyme and lysed by sonication using 10-s pulses with 50 W for a total time of 10

min. Lysates were clarified by centrifugation at 50,000$g$ for 1 h at 4°C, and the supernatant was used for affinity purification on HisTrap FF columns (GE Healthcare) according to the manufacturer's instructions. For all constructs, the 6His-Smt3 tag was removed by incubation with Senp1$^{415-644}$ and concurrent dialysis in binding buffer. The liberated affinity tag and the His-tagged Senp1 protease were removed by a second round of affinity purification with HisTrap FF columns (GE Healthcare). All proteins were purified with a final size exclusion chromatography (HiLoad 16/600 Superdex 75 or 200 pg) in 20 mM Tris, pH 7.5, 150 mM NaCl, and 2 mM DTT and concentrated using VIVASPIN 20 Columns (Sartorius), flash-frozen in liquid nitrogen, and stored at –80°C.

All proteins used in the acetyltransferase activity assay were expressed in *E. coli* BL21 (DE3). For protein expression of the non-acetylated proteins, bacterial cells were grown to an OD$_{600}$ of 0.6 (37°C; 160 rpm). Afterward, protein expression was induced by addition of 400 $\mu$M of IPTG and was conducted overnight (18°C; 160 rpm). For the lysine-acetylated His$_6$-MEK AcK210, the pRSF-Duet1 vector encoded for the synthetically evolved acetyl-lysyl-tRNA$_{CUA}$-synthetase/tRNA$_{CUA}$ pair derived from the PylRS/tRNA$_{CUA}$ system from *Methanosarcina barkeri* was used as described earlier (41). In short, the cells were grown to an OD$_{600}$ of 0.5, and 10 mM acetyl-L-lysine and 20 mM nicotinamide were added to the culture. Subsequently, the culture was grown for additional 30 min (37°C, 160 rpm). Afterward, protein expression was induced by addition of 200 $\mu$M IPTG and was performed overnight (18°C, 160 rpm). After expression, the cells were harvested by centrifugation (4,000$g$, 20 min) and resuspended in resuspension buffer (RP1 for pOPIN/pGEX-4T5 expressions: 50 mM Tris/HCl pH 7.4, 100 mM NaCl, 5 mM MgCl$_2$, 2 mM $\beta$–mercaptoethanol, and 100 $\mu$M Pefabloc; RP2 for pRSF-Duet1 expressions: as RP1 plus 20 mM imidazole, for His$_6$-MEK6 AcK210 RP1 plus 20 mM imidazole and 20 mM nicotinamide to inhibit *E. coli* CobB deacetylase). Cell lysis was performed by sonication, and the soluble fraction (50,000$g$, 45 min) was applied to the equilibrated Ni-NTA affinity-chromatography material or the glutathione (GSH)-column, respectively. Washing was performed with washing buffer (WB1 for GSH-column: 50 mM Tris/HCl pH 7.4, 500 mM NaCl, 5 mM MgCl$_2$, and 2 mM $\beta$-mercaptoethanol; WB2 for Ni-NTA material: as WB1 plus 20 mM imidazole). For GST-LPG1949$_{1-291}$, the fusion protein was cleaved by TEV protease cleavage on the column overnight at 4°C. The His$_6$-SUMO-tag was removed by SENP1 cleavage for His$_6$-SUMO-LPG1949 full-length following elution from the Ni-NTA material using 500 mM imidazole and during dialysis. The cleaved His$_6$-SUMO tag and uncleaved His$_6$-SUMO-LPG1949 protein was subsequently removed by Ni-NTA affinity chromatography. For the His-MEK6 and acetylated His$_6$-MEK6 AcK210, the protein was applied to Ni-NTA material, eluted by a step gradient up to 500 mM imidazole. For all proteins, the protein of interest containing fractions were concentrated using an Amicon ultrafiltration unit and applied to a suitable size-exclusion chromatography column (for LPG1-1949 full-length and His$_6$-MEK6/His$_6$-MEK6 AcK210: S200 16/600 or S200 10/300; for LPG1-1949$_{1-291}$: S75 16/600; GE Healthcare). Afterward, the fractions containing the protein of interest were concentrated, shock-frozen in liquid nitrogen, and stored at –80°C. Protein concentrations were determined using the absorption at 280 nm (A$_{280}$) using the proteins' extinction coefficients derived from their sequences.

## Synthesis of activity-based probes

All Ub/UbL activity-based probes used in this study were expressed as C-terminal intein fusion proteins as described previously (42). The fusion proteins were affinity purified in buffer A (20 mM Hepes, 50 mM sodium acetate, pH 6.5, 75 mM NaCl) from clarified lysates using Chitin Resin (New England Biolabs) following the manufacturer's protocol. On-bead cleavage was performed by incubation with cleavage buffer (buffer A containing 100 mM MesNa [sodium 2-mercaptoethanesulfonate]) for 24 h at RT. The resin was washed extensively with buffer A and the pooled fractions were concentrated and subjected to size exclusion chromatography (HiLoad 16/600 Superdex 75) with buffer A. To synthesize Ub/UbL-PA, 300 $\mu$M Ub/UbL-MesNa were reacted with 600 $\mu$M propargylamine hydrochloride (Sigma-Aldrich) in buffer A containing 150 mM NaOH for 3 h at RT. Unreacted propargylamine was removed by size exclusion chromatography and Ub/UbL-PA was concentrated using VIVASPIN 20 Columns (3 kD cutoff; Sartorius), flash-frozen, and stored at −80°C.

## Chain generation

Met1-linked di-ubiquitin was expressed as a linear fusion protein and purified by ion exchange chromatography and size exclusion chromatography. K11-, K48-, and K63-linked ubiquitin chains were enzymatically assembled using UBE2S$\Delta$C (K11), CDC34 (K48), and Ubc13/UBE2V1 (K63) as previously described (43, 44). In brief, ubiquitin chains were generated by incubation of 1 $\mu$M E1, 25 $\mu$M of the respective E2 and 2 mM ubiquitin in reaction buffer (10 mM ATP, 40 mM Tris [pH 7.5], 10 mM MgCl$_2$, 1 mM DTT) for 18 h at RT. The reaction was stopped by 20-fold dilution in 50 mM sodium acetate (pH 4.5) and chains of different lengths were separated by cation exchange using a Resource S column (GE Healthcare). Elution of different chain lengths was achieved with a gradient from 0 to 600 mM NaCl.

## AMC assays

Activity assays of DUBs against AMC-labeled Ub/UbL substrates were performed using reaction buffer (150 mM NaCl, 20 mM Tris, pH 7.5, and 10 mM DTT), 1 $\mu$M DUBs, 5 $\mu$M Nedd8-AMC (ENZO Life Sciences GmbH), or 5 $\mu$M Ub-AMC (UbiQ-Bio). The reaction was performed in black 96-well plates (Corning) at 30°C and released fluorescence was measured using the Infinite F200 Pro plate reader (Tecan) equipped for an excitation wavelength of 360 nm and an emission wavelength of 465 nm. The presented results are means of three independent cleavage assays.

## Activity-based probe assays

DUBs were prediluted to 2× concentration (10 $\mu$M) in reaction buffer (20 mM Tris, pH 7.5, 150 mM NaCl, and 10 mM DTT) and 1:1 combined with 100 $\mu$M Ub-PA, NEDD8-PA, ISG15$^{79–156}$-PA, and SUMO1-PA or SUMO3-PA. After 3-h incubation at RT, the reaction was stopped by addition of 2× Laemmli buffer, resolved by SDS–PAGE, and Coomassie stained. At least two technical replicates were performed and one representative gel is shown.

## Ubiquitin chain cleavage

DUBs were preincubated in 150 mM NaCl, 20 mM Tris, pH 7, and 10 mM DTT for 10 min. The cleavage was performed for the indicated time points with 5 $\mu$M DUBs and either 25 $\mu$M di-ubiquitin (K11, K48, K63, and M1 synthesized as described above, others from Boston Biochem) or 5 $\mu$M tetra-ubiquitin (Boston Biochem) at RT. After the indicated time points, the reaction was stopped with 2× Laemmli buffer, resolved by SDS–PAGE, and Coomassie stained. At least two technical replicates were performed and one representative gel is shown.

## Acetyltransferase (AcT) activity of LPG1949

The acetyltransferase activity of LPG1949 was assessed using immunoblotting as a readout. The activity assays were performed in a total volume of 20 $\mu$l in assay buffer (50 mM Tris/HCl, pH 7.4, 100 mM NaCl, 5 mM MgCl$_2$, and 2 mM $\beta$-mercaptoethanol). LPG1949 full-length and LPG1949$^{1–291}$ were used at a final concentration of 4 $\mu$M. The proteins were analyzed for autoacetylation activity. In addition, His$_6$-MEK6 protein was used as a substrate at a final concentration of 10 $\mu$M. To assess the possible influence of inositol-hexaphosphate (IP6) on AcT activity (45), samples were conducted containing 1 mM IP6. The acetyl-group donor molecule for AcT activity, acetyl-CoA, was used at a final concentration of 1 mM. The reactions were incubated for 3 h at 37°C. Afterward, the reactions were stopped by addition of 14 $\mu$l H$_2$O and 16 $\mu$M 3× sample buffer and incubation for 10 min at 90°C. The reactions were analyzed using immunoblotting following a standard protocol. As a primary antibody, a rabbit anti–acetyl-lysine antibody was used (ab21623; Abcam). Detection was performed by chemiluminescence using a goat antirabbit antibody (ab205718; Abcam) coupled to an HRP and ECL reagent as a substrate for HRP. As a positive control, acetylated His$_6$-MEK6 AcK210 was used, and Ponceau S staining of the membrane was performed for a loading control. At least two technical replicates were performed and one representative gel is shown.

# Supplementary Information

# Acknowledgements

We would like to thank Axel Hamprecht for *Legionella* DNA samples and David Komander for reagents and helpful discussions. Work in the Hofmann lab has been supported by grants from the Deutsche Forschungsgemeinschaft (CRC670 and CRC1403). M Lammers is supported by a grant from the Deutsche Forschungsgemeinschaft (LA 2984/5-1).

## Author Contributions

T Hermanns: investigation and performed most experiments.

I Woiwode: investigation and generated and performed assays with UbL probes.

RFM Guerreiro: software, investigation, visualization, and performed the protease clustering and some bioinformatical analyses.

R Vogt: investigation and performed the acetyltransferase assays.

M Lammers: investigation and performed the acetyltransferase assays.

K Hofmann: conceptualization, supervision, funding acquisition, investigation, and writing—original draft and initiated and supervised the study and performed bioinformatical and structural analyses.

## Conflict of Interest Statement

The authors declare that they have no conflict of interest.

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
