## [Reviewer comments · Life Science Alliance]

Life Science Alliance

An evolutionary approach to systematic discovery of novel deubiquitinases, applied to *Legionella*

Thomas Hermanns, Ilka Woiwode, Ricardo Guerreiro, Robert Vogt, Michael Lammers, and Kay Hofmann

DOI: <https://doi.org/10.26508/lsa.202000838>

Corresponding author(s): Kay Hofmann, University of Cologne

Review Timeline:	Submission Date:	2020-07-02
	Editorial Decision:	2020-07-10
	Revision Received:	2020-07-13
	Accepted:	2020-07-15

Transaction Report:

Please note that the manuscript was previously reviewed at another journal and the reports were taken into account in the decision-making process at Life Science Alliance.

Reviewer #1 (General assessment and major comments (Required)):

The paper covers an interesting topic, is well written and presented. The authors combine bioinformatic, structural and biochemical data for the identification and characterization of cysteine-DUBs and their distinguishing features compared to other cysteine proteases. Their data confirms the function of two previously studied DUBs, Ipg1148 (*Legionella* ubiquitin-specific protease A (Urbanus et al., Mol Syst Biol 2016); PDB 5DGG) and Ipg1621 (Deubiquitinase Ceg23 (Ma et al., JBC 2020); PDB 6KS5). Furthermore, the paper extends our knowledge by providing an extensive analysis of cysteine protease sequence relationships and by studying the importance of a conserved aromatic residue for DUB activity.

Substantive Concerns

(1) Previous work on the two DUBs Ipg1148 and Ipg1621 is not discussed objectively or credited appropriately. The study on Ipg1148 is mentioned on p.9, concluding that "the DUB activity and specificity of Ipg1148 were not characterized in any detail". In fact, this study reports the structure of Ipg1148 ("*Legionella* ubiquitin-specific protease A"; PDB 5DGG), shows its *in vitro* and *in vivo* DUB activity and identifies its physiological substrate LegC3.

We are sorry to hear that this referee feels that we have not credited the publications of Ipg1148 and Ipg1621 appropriately. The (Urbanus et al 2016) work on Ipg1148 came to our attention long after we performed our own characterization of this enzyme: neither PUBMED searches for Ipg1148 nor for *Legionella* deubiquitinases will find this paper, since its focus is on a completely different topic. What Urbanus et al do show is that Ipg1148 is able to cleave the model substrate Ub-AMC and is able to remove ubiquitin from the LegC3 protein. No attempts have been made to test if Ipg1148 acts specifically on LegC3 or if it can cleave other targets such as ubiquitin chains. In addition, no attempts for elucidating the linkage specificity of Ipg1148 have been described. In our opinion, these are the most crucial components for a DUB characterization. We are referring to the Urbanus et al paper on several occasions and compare our results to those provided in this publication. To avoid the impression of belittling this publication, we have toned down our original statement that "the DUB activity and specificity of Ipg1148 were not characterized in any detail".

The (very recent) publication on Ipg1621, is only mentioned in the discussion "in full agreement with the catalytic properties reported here". The crystal structures or physiological roles of both proteins are not mentioned at all. This is particularly astonishing, as the crystal structures could provide important insights on the role of the "aromatic gatekeeper motif" and affect the interpretation of the results depicted in Figure 5. Overall, the authors' description of the current knowledge on Ipg1148 and Ipg1621 is highly misleading, and implementing the key results of these studies is essential to give the reader a clear picture.

The publication of Ipg1621 appeared during the proofreading stage of our own manuscript. To give proper credit to this publication, we had added it to the discussion section of our manuscript and stated that the results described there are in full agreement with our own results. We did not have the chance to analyze the published structure of Ipg1621 in our original manuscript, because the structure became available only after the date of our submission.

In the revised version, we added a discussion of the aromatic gatekeeper motif in Ipg1148 and Ipg1621, supported by the newly created supplementary figure S3. Since both new structures cover only the apo-form of the enzyme without a bound or covalently attached ubiquitin, we visualized the likely orientation of the ubiquitin C-terminus in figure S3 by using related structures (XopD for Ipg1148 and OTUD2 for Ipg1621)

(2) The relevance of the "aromatic gatekeeper motif" is not substantiated sufficiently. While the alignment in Figure 2 suggests that most DUBs feature an aromatic residue at this position, no data is presented on other cysteine proteases. Thus, the role of this "motif" as discriminating feature between cysteine-DUBs and other cysteine proteases is not supported.

This is a very good point. We do have data on the presence (or rather: absence) of the aromatic motif in almost all other cysteine protease classes. We have added a paragraph to the revised manuscript, which describes the statistical significance and enrichment factor. The details of this analysis are provided as (new) Suppl. Table 1.

Furthermore, it may be beneficial to include structural data as in Figure S2 in the main article, possibly with electron densities for the aromatic side chain and the hypothetical C β to show potential steric clashes.

We have now promoted the former supplementary figure S2 to become main figure 2. To provide further support for the claims on the importance of the aromatic motif, we have now performed mutagenesis experiments on the corresponding motif in several other DUB classes (USP, UCH, Josephins). The new data have been added to figure 6 (originally: figure 5) as new panels f-h. The results are described in a newly added paragraph.

Reviewer #1 (Minor Comments):

(1) In the abstract and the introduction (p. 3), the question is raised whether DUB families have a common evolutionary ancestry or evolved ubiquitin-specificity independently, but it is not addressed within the results or discussion section. An important point in this respect is that the CA and CE clans, which include all currently known DUBs, are suggested to be related by circular permutation (according to MEROPS/SCOP databases).

This aspect is discussed in a new paragraph, which we have added to the discussion section.

(2) The color scheme in Figure 1 and 2 could be reworked. Red and blue indicate clan assignment in Figure 1, but function in Figure 2. I suggest to show clan assignment and function in both figures with a consistent color scheme/style.

Figures 1 and 2 (now figure 3) have a very different purpose, which – in our opinion – justifies the use of two different coloring schemes. The purpose of figure 1 is to give an overview of the families and clans represented in MEROPS and to show which of these families are related to each other, even including several inter-clan relationships. Here, it is important to visually highlight the clan assignments, which requires 17 different colors. By contrast, the purpose of figure 3 (originally figure 2) is to highlight relationships between DUB families, UBL-protease families and candidate families from *Legionella* – irrespective of the clan assignment. In our opinion, the focus of this figure should be on the clear discrimination of the functional categories – without giving the impression that the ‘functional colors’ have any connection to the ‘clan coloring’ of figure 1.

(3) One major conclusion from Figure 1 is that most viral, bacterial and eukaryotic families are connected (p. 6), but Figure 1 depicts no taxonomic information. Overall, the relationships between cysteine-DUBs and/or other cysteine proteases would be easier to appreciate if the DUB classes were indicated (e.g. UCH instead of just c12).

The purpose of figure 1 is to give an overview of the connections between MEROPS clans, without really focusing on DUBs. For that reason, we show the MEROPS family numbers. Figure 2A is actually a subset of Figure 1, using a different coloring- and labeling scheme. Since the focus of figure 3a (originally: 2a) is on the DUBs, the network node corresponding to MEROPS C12 is here labeled UCH. We would strongly prefer to leave the colors and labels of figure 1 unchanged. Note that in the supplementary material, we also provide the complete Cytoscape network, on which figures 1 and 3 are based. Using the open source program Cytoscape, different coloring- and naming-schemes for these figures can be easily explored.

(4) In Figure S1/S2, the ubiquitin moiety should always be shown in the same orientation (and at the same zoom level) to facilitate comparability.

We have tried to comply with this request, but feel that all attempts to unify orientation and zoom factor lead to a loss of clarity. When fixing the orientation, there are always some structures where important features come to lie on top of each other, occluding useful information. In the new figure 2 (originally S1), we keep the zoom factor constant and allow only minor deviations from a constant orientation – just enough to prevent obstructions. In the new Suppl. Figure S1 (originally S2), we kept using a somewhat different zoom factor, since the RLRGG-recognition in some families is much more ‘compact’ than in others. We feel that a constant zoom level would deteriorate the readability of this figure.

(5) The nature of the "activity-based Ub probe" / "Ub-PA" should be explained (p.9). Propargylamide

(PA) is in fact an alkyne-based inhibitor, that reacts with the active site cysteine in a very different manner compared to an isopeptide-bonded ubiquitin. A modification with Ub-PA shows that the DUB binds ubiquitin and features a particularly nucleophilic side-chain. This may suggest DUB activity, but is no proof for it. Knowledge of the chemical nature of PA is also essential to understand the modified active sites in Figure S2.

We have added an explanatory sentence on Ub-PA function. We would like to emphasize that we never considered a reaction with a PA-based probe sufficient for claiming a DUB activity. In all cases, we also provide data on chain cleavage, including different linkage types.

(6) Likewise, it should be explained what IP6 is and why it is used (Figure 3F).

IP6 is inositol hexaphosphate, which often stimulates bacterial acetyltransferases and other effectors acting on membrane-associated components. We have added an explanation and a literature reference.

(7) The gene numbers on p.9 (lpg0027 and lpg2959) may be incorrect (lpg2529 and lpg0227 instead?).

We have corrected this mistake in the revised manuscript.

(8) Figure 4e: The bands for the lpg1148 1-305 variant are much more intense than those for the 123-305 or 1-305 C252A variants, indicating that different enzyme concentrations were used. Please clarify.

Equal protein concentrations had been used in this experiment. Since the band of the lpg1148 1-305 variant looked indeed more intense than expected, we have re-purified the enzyme, determined the enzyme concentration, and repeated the experiment. The new data show a more uniform intensity distribution (Figure 5e, originally 4e) while confirming the original conclusion.

(9) Figure 5c: It is unclear, why Ub6+ is used as a substrate, although all previous assays with lpg1621 (Fig. 4a-c) as well as the analogous lpg1148 experiment (Fig. 5b) were conducted with di- and tetraubiquitin.

There was no conceptual reason for using the Ub6+ chains in this experiment. At that time point, we had only these chains left and used them because for comparing the mutant activity, the chain length shouldn't make a difference. We have now repeated the experiment with di-ubiquitin and show the result in Suppl. Figure S3c. As expected, the conclusions remain unchanged.

(10) The claim that DALI fails to detect similarities between RavD and other cysteine proteases appears to be incorrect. Running a search with RavD (PDB 6NII) yields a great variety of cysteine proteases and DUBs, with 1UKF (AvrPphB) being the top hit.

Our original manuscript claimed that DALI doesn't find significant relationships to other DUBs and cysteine proteases, based on the observation that all Z-scores are below 6 (six-sigma criterion). I became aware that indeed most researchers also consider DALI scores below 6 as significant. For this reason, we have rephrased this statement to make it more precise.

Reviewer #2 (General assessment and major comments (Required)):

The significant aspect of this work is the complete chain of analysis starting from a thorough investigation of the papain-like thiol peptidases and culminating in wet-lab results testing the hypothesis in the endo-parasitic bacterium Legionella, which secretes a diverse set of peptidase effectors into its host cell. While a large number of thiol peptidases are known from viruses and pathogenic/symbiotic bacteria, their role as DUBs has been difficult to establish purely based on sequence analysis unless accompanying domains offer contextual information. Even then the prediction leaves some room for uncertainty because the accompanying domains might merely indicate interaction with Ub and not DUB activity. The primary finding of Hermanns et al is that a gatekeeper aromatic residue, found immediately after the catalytic H, provides the selectivity for ubiquitin. They show using the Legionella peptidases that this is indeed a good predictor though not a sufficient one given that one of the test cases proved to be an acyltransferase rather than an

isopeptidase.

An important issue is that the several papain-like fold peptidases with a comparable aromatic residue are also found in the papain-like fold toxin-1 family (Pfam: Pf15644; apparently not in MEROPS) some of which have been shown to possess glutamine amidase activity where they hydrolyze the amido group of glutamine to generate glutamate. Given that at least one bacterial pathogen interferes with the host ubiquitin conjugation via secretion of a such an enzyme and deamidation of a Q residue, could this be an activity that also exists among the effectors of legionellae? It would be good to see some computational or wet-lab considerations to see why such an activity was not considered by the authors for their candidates.

Our analysis was originally built on MEROPS, assuming that this database would cover all known cysteine protease families. Since we found several gaps in the MEROPS coverage, we manually added several missing protease families to our analysis (yellow boxes in figure 1). In the original manuscript, the PF15644 "papain-fold toxin" family was not included because these enzymes are glutamine deamidases and we are not aware of any family member with protease activity. In the revised manuscript, we have added this family (label: pftox1) to the analysis; it is shown as a connected box in figure 1 and supplementary figure S2 and is also counted in the statistical analysis (Supplementary table 1). While single members of the papain-like fold toxin-1 (pftox1) family do have an aromatic residue after the catalytic histidine, this is not a conserved property of the family. For this reason, the pftox1 family does not fulfill the requirements of a 'conserved aromatic motif' and is thus listed in the 3rd data row of Suppl. Table 1.

It would also be useful to survey certain proteomes of bacteria that are unlikely to be eukaryotic pathogens/symbionts as a "negative control" to show that thiol peptidases with the discriminatory aromatic residue is absent in them.

We have done this on several such bacterial species (e.g. *Bacillus subtilis*) and don't find protease families with the conserved gatekeeper motif. There are, however, some innocuous bacterial species that have retained DUB-like genes from their pathogenic ancestry. One example is ElaD, which is found in all enterobacteria irrespective of their pathogenicity – including laboratory strain *Escherichia coli*. We have discussed this situation in a recent review. (<https://www.ncbi.nlm.nih.gov/pubmed/31845741>)

Reviewer #2 (Minor Comments):

The distinction between DUBs and DUBLs is not presented too clearly in the introduction. We have added an explanation to the introduction section and have added one literature reference.

Do the authors imply the monophyly of all DUBs or convergent evolution of the aromatic position? Did DUBLs simply did not evolve such a position while being more related to certain DUBs in reality? Monophyly of all cysteine-DUBs (and UBL proteases) would be bold claim, to which I am sympathetic, but which is not (yet) sufficiently supported by data. Most UBL-proteases (e.g. for SUMO, NEDD8, ISG15) are clearly part of DUB families that contain the gatekeeper motif. By contrast, the protease for UFM1 and ATG8 are related to the deubiquitinase ZUFSP/ZUP1 and all of them lack the aromatic motif. Based on our analysis, I consider it likely that the loss of the aromatic residue is a derived feature that allowed this family to cleave UBL modifiers not ending on GlyGly. We can include this discussion in the revised manuscript.

It would be useful to illustrate the overall topology with the positions of the catalytic residues and aromatic gatekeeper for the DUB classes either in the supplement or as a figure.

We have explored topology diagrams (in the style generated by pro-origami) but did not really find them helpful. Instead, we have added a paragraph to the discussion where the topology is discussed in the context of the conserved papain fold.

The original term used for the peptidase of the seventh DUB class is the JAB superfamily. It is suggested that the same be used.

The name "JAB" was used in one of the original papers (Verma et al 2002), a simultaneous paper

used "MPN+" (Maytal-Kivity et al 2002). The current ubiquitin literature consistently uses the acronym JAMM (for "Jab/MPN domain-associated metalloisopeptidase"), which combines the two original names. We have added the explanation of the name and have also added references to the JAB-paper and the MPN-paper.

The thiol peptidase DUBs and DUBLs do share a common structural scaffold even if it might show great elaboration, decoration with inserts and circular permutation in several cases. This is the papain-like fold which can be clearly distinguished from all other thiol peptidases (even recognized in the original SCOP database), for example the caspase-like fold. There is no mention of this in the article. This would help better orient a reader as to what part of the thiol peptidase "universe" these enzymes come from.

This aspect is covered in a new paragraph added to the discussion section.

Reviewer #3 (General assessment and major comments (Required)):

This is another excellent contribution from the lab of the leading bioinformatics expert of the ubiquitin field following on from their most recent description of the 7th clade of DUBs (ZUFSP) represented in mammals. The study identifies firstly a key sequence determinant that is common to nearly all DUBs and may help uncover new candidate DUBs, secondly provides experimental evidence for the functional significance of these conserved features and thirdly provides further evidence for this new insight by using it to conduct a genome wide predictive search that leads to the identification of as yet uncharacterised DUBs in the Legionella pneumophila genome. Importantly, the analysis and identification of new more distantly related DUBs reveals links between hitherto unconnected families suggesting the potential diversification arose from a common ancestor. This is another important stepping stone in gaining a comprehensive overview of a family that harbours many pathogenically and translationally relevant enzymes.

It contributes to our understanding of their catalytic site architecture, identifies new potentially druggable members associated with a pathogen, and provides key insights in the phylogenetic diversity of this protease family.

I fully support the publication of this interesting manuscript and have only minor comments and suggestions that are listed below.

Reviewer #3 (Minor Comments):

Page 3: " where the JAMM protease Rpn11 recycles ubiquitin from degradation targets" - it may be worth pointing out here that Rpn11 here works in conjunction with an inactive member of the family. We have added this information to the introduction section.

Page 4: "Also included are some MEROPS serine protease families of the mixed clan PA, which are known to be related to cysteine proteases" - are these marked in any particular way in the figure? They have family names starting with an "S", such as S07, S29 etc. We have added this explanation to the legend of figure 1.

P7: "Interestingly, the aromatic motif is absent from proteases cleaving UFM1 (C78) and ATG8 (C54), two modifiers that do not possess glycine at the penultimate position."

It would be helpful in view of the later discussion of G75V mutations, to indicate here what the residue is standing in for G75 in those UBLs that do have a substitution in this position.

Unlike other modifiers all ending on GG, UFM1 ends on VG, while most ATG8 forms end on FG. This information has been added to the revised manuscript.

Figure 3B - the size of the font should be increased

We have increased the font size in this figure, and also in figures 5b, 4f and 4h.

For the discussion:

Igp1949: is the aromatic residue nevertheless important for this enzyme? or is this an evolutionary vestige? Is it not in principle possible that Igp1949 is in fact another example of a dual activity enzyme but with a cryptic DUB activity that may require activation for example through a post-translational modification? This paper also provides a further interesting point for discussion (Biochem J. 2008 Nov 1;415(3):367-75).

We consider it to be an evolutionary vestige, but cannot formally exclude a cryptic DUB activity. We have added the mentioned paper to a newly added paragraph in the discussion section.

July 10, 2020

RE: Life Science Alliance Manuscript #LSA-2020-00838

Dr. Kay Hofmann
University of Cologne
Zülpicher Str 47a
Cologne 50674
Germany

Dear Dr. Hofmann,

Thank you for submitting your revised manuscript entitled "An evolutionary approach to systematic discovery of novel deubiquitinases, applied to Legionella" that was reviewed at another journal. We would be happy to publish your paper in Life Science Alliance pending final revisions necessary to meet our formatting guidelines.

- please include discussion in your PBP response on the minor issue 'distinction between DUBs and DUBLs' in the manuscript discussion, as suggested by Ref#2
- please look at our Manuscript Preparation guidelines and separate your manuscript sections accordingly
- please upload your main and supplementary figures as single files
- please add your figure legends as a separate section in the main manuscript text
- please provide your tables as a separate file in editable docx or excel format

A. FINAL FILES:

-- Summary blurb (enter in submission system): A short text summarizing in a single sentence the study (max. 200 characters including spaces). This text is used in conjunction with the titles of

papers, hence should be informative and complementary to the title. It should describe the context and significance of the findings for a general readership; it should be written in the present tense and refer to the work in the third person. Author names should not be mentioned.

B. MANUSCRIPT ORGANIZATION AND FORMATTING:

Sincerely,

Reilly Lorenz
Editorial Office Life Science Alliance
Meyerhofstr. 1
69117 Heidelberg, Germany
t +49 6221 8891 414
e contact@life-science-alliance.org
www.life-science-alliance.org

July 15, 2020

RE: Life Science Alliance Manuscript #LSA-2020-00838R

Dr. Kay Hofmann
University of Cologne
Zùlpicher Str 47a
Cologne 50674
Germany

Dear Dr. Hofmann,

Thank you for submitting your Research Article entitled "An evolutionary approach to systematic discovery of novel deubiquitinases, applied to Legionella". It is a pleasure to let you know that your manuscript is now accepted for publication in Life Science Alliance. Congratulations on this interesting work.

DISTRIBUTION OF MATERIALS:

Again, congratulations on a very nice paper. I hope you found the review process to be constructive and are pleased with how the manuscript was handled editorially. We look forward to future exciting submissions from your lab.

Sincerely,

Reilly Lorenz
Editorial Office Life Science Alliance
Meyerhofstr. 1
69117 Heidelberg, Germany
t +49 6221 8891 414
e contact@life-science-alliance.org
www.life-science-alliance.org